# Differential Expression of LLT1, SLAM Receptors CS1 and 2B4 and NCR Receptors NKp46 and NKp30 in Pediatric Acute Lymphoblastic Leukemia (ALL)

**DOI:** 10.3390/ijms24043860

**Published:** 2023-02-15

**Authors:** Sheila B. Powers, Nourhan G. Ahmed, Roslin Jose, Marissa Brezgiel, Subhash Aryal, W. Paul Bowman, Porunelloor A. Mathew, Stephen O. Mathew

**Affiliations:** 1Department of Microbiology, Immunology and Genetics, University of North Texas Health Science Center, Fort Worth, TX 76107, USA; 2Texas College of Osteopathic Medicine, University of North Texas Health Science Center, Fort Worth, TX 76107, USA; 3School of Nursing, University of Pennsylvania, Philadelphia, PA 19104, USA; 4Cook Children’s Medical Center, 801 7th Avenue, Fort Worth, TX 76104, USA

**Keywords:** acute lymphoblastic leukemia, natural killer cell, 2B4, LLT1, CS1, NKp30, NKp46

## Abstract

Acute lymphoblastic leukemia (ALL) represents the most common pediatric cancer. Most patients (85%) develop B-cell ALL; however, T-cell ALL tends to be more aggressive. We have previously identified 2B4 (SLAMF4), CS1 (SLAMF7) and LLT1 (CLEC2D) that can activate or inhibit NK cells upon the interaction with their ligands. In this study, the expression of 2B4, CS1, LLT1, NKp30 and NKp46 was determined. The expression profiles of these immune receptors were analyzed in the peripheral blood mononuclear cells of B-ALL and T-ALL subjects by single-cell RNA sequencing data obtained from the St. Jude PeCan data portal that showed increased expression of LLT1 in B-ALL and T-ALL subjects. Whole blood was collected from 42 pediatric ALL subjects at diagnosis and post-induction chemotherapy and 20 healthy subjects, and expression was determined at the mRNA and cell surface protein level. A significant increase in cell surface LLT1 expression in T cells, monocytes and NK cells was observed. Increased expression of CS1 and NKp46 was observed on monocytes of ALL subjects at diagnosis. A decrease of LLT1, 2B4, CS1 and NKp46 on T cells of ALL subjects was also observed post-induction chemotherapy. Furthermore, mRNA data showed altered expression of receptors in ALL subjects pre- and post-induction chemotherapy treatment. The results indicate that the differential expression of the receptors/ligand may play a role in the T-cell- and NK-cell-mediated immune surveillance of pediatric ALL.

## 1. Introduction

Acute lymphoblastic leukemia (ALL) is an aggressive hematologic malignancy characterized by blood and bone marrow infiltration by malignant lymphoblasts. According to the NCI, ALL is the most common cancer in the US among children aged 0–14. However, when it happens to adults, it is more aggressive and will most probably relapse even after complete remission [1,2,3,4,5]. Since leukemic cells are a defective version of lymphoblasts, they leave the body anemic and more susceptible to infections that would eventually cause death if left untreated. The first in line treatments for ALL remain a combination of several chemotherapy agents, namely vincristine, glucocorticoids and an anthracycline as doxorubicin followed by allogenic stem cell transplantation for high-risk subjects. Some chemotherapy regimens involve asparaginase as well [1,6,7,8,9,10,11].

Children and adolescents with ALL treated with pediatric regimens have a high long-term remission rate that can reach 85–94% [1,3,6,7,8,12,13,14]. However, those of them who relapse have a much lower survival rate of 30–50% after the first relapse, since most relapses occur in the bone marrow with or without the involvement of a secondary site as the CNS or testes [1,14]. The strategy of immunotherapy has amassed great interest in recent years. Immunotherapy is specific in the targeting of the disease so that healthy cells are typically not destroyed. Because of this, the side effects are minimal, and this therapy can be utilized in conjunction with conventional therapies [6,9,15,16,17,18]. Developing an immunotherapy relies heavily on understanding the mechanism and cellular environment of the disease.

Previous studies have shown that ALL of the B-cell lineage is particularly resistant to killing by natural killer (NK) cells by impairing their activation and lowering their immune surveillance potential [19]. NK cells are immune cells of the innate immune system that target virally infected and cancer cells. NK cells’ activation mechanism works differently from that of T cells since they directly interact with the target cells without the need for antigen presentation [20,21,22]. NK cells are typically classified under innate immunity, although they do have characteristics of adaptive immunity, such as clonal expansion of antigen-specific NK cells, longevity and a more vigorous response upon reinfection, indicating a memory-like response [23,24,25]. Consequently, NK cells have garnered significant interest in cancer research.

Currently, NK-cell dysfunction has been linked to many cancers, playing at least some part in onset or continuance of the cancer [21,26,27]. Specifically, the NK cells do not appear to achieve the proper activating signals through their receptors. In this study, we first explored the expression of immune receptors by analyzing RNA-seq data from pediatric ALL subjects (B-ALL and T-ALL) from the St. Jude Pediatric Cancer genomic database. Based on our preliminary analysis, we selected five immune receptors for further evaluation in a cohort of ALL and healthy subjects. Three of these receptors, 2B4 (CD244), CS1 (CRACC, CD319) and LLT1 (CLEC2D), were cloned in our laboratory [28,29,30] and have been shown to play a role in other cancers and diseases, such as Systemic Lupus Erythematosus (SLE), prostate cancer and Triple-Negative Breast Cancer (TNBC), as well as other leukemias [31,32,33,34,35,36]. The other two receptors, NKp30 and NKp46, are well-known activating receptors of NK cells against cancer cells and have already been identified as receptors of interest in ALL [37,38,39]. Overall, this study provides new insight to the expression of these receptors and their role in the immune dysregulation in childhood ALL.

## 2. Results

### 2.1. RNA Seq Expression Analysis of Immune Receptors in B-ALL and T-ALL Subjects

The expression profiles of immune receptors 2B4 (*CD244)*, CS1 (*SLAMF7)*, LLT1 (*CLEC2D)*, NKp46 (*NCR1*) and NKp30 (*NCR3)* were analyzed in the peripheral blood mononuclear cells of B-ALL and T-ALL subjects by single-cell RNA sequencing data obtained from the St. Jude PeCan data portal. In both B-ALL subjects (*n* = 729) and T-ALL subjects (*n* = 313), LLT1 was significantly overexpressed with fragments per kilobase of exon per million mapped fragments (FPKM) values of 32.79 and 8.29, respectively (Figure 1). When compared with all other pediatric cancers (FPKM = 14.69), *LLT1* expression was significantly overexpressed in B-ALL (FPKM = 32.79) subjects, whereas it was decreased in T-ALL (FPKM = 8.3) subjects. CS1 (FPKM = 1.9) and NKp30 (FPKM = 1.69) were also overexpressed in B-ALL as compared to other cancers (FPKM = 1.54) and (FPKM = 1.33), respectively. Additionally, 2B4 and NKp30 expression was significantly elevated in T-ALL subjects with FPKM values of 7.54 and 2.34, respectively. It is also worth mentioning that NKp46 expression was not significantly different in B- (FPKM = 0.69) and T-ALL (FPKM = 0.61) subjects from other types of cancers (FPKM = 0.71) (Figure 1).

### 2.2. mRNA Expression of Immune Receptors in Pediatric ALL Subjects

To assess the gene expression in the pediatric ALL subjects recruited in our cohort study, we isolated mRNA from the PBMC of ALL subjects (*n* = 42) at diagnosis (day 0) and at the end of induction chemotherapy treatment (day 29) as well as in healthy subjects (*n* = 20). qRT-PCR analysis was conducted to compare the expression of the immune receptors.

The qRT-PCR data showed an overexpression of 2B4 (*CD244*), CS1 (*SLAMF7*), LLT1 (*CLEC2D*) and NKp30 (*NCR3*) in ALL subjects at diagnosis as compared to healthy subjects, although it was not statistically significant (*p* > 0.05). There was a statistically significant decrease in NKp46 (*NCR1*) expression in ALL subjects at diagnosis (*p* < 0.005) and at conclusion of induction chemotherapy (*p* < 0.05) when compared to the healthy subjects, as shown in Figure 2. On the other hand, upon the conclusion of the induction chemotherapy (day 29) for the ALL subjects, there was differential expression of genes for each of the receptors. 2B4 (*CD244*) and LLT1 (*CLEC2D*) were both downregulated as compared to their expression at diagnosis, but the expression was higher than in healthy subjects. CS1 expression after induction chemotherapy decreased by three folds compared to its expression at diagnosis, whereas there was an increase in the expression of NKp30 and NKp46 after treatment as compared to their expression at diagnosis (Figure 2).

### 2.3. Differential Cell Surface Expression of Immune Receptors on Monocytes (CD14+ Cells) in ALL Subjects

To assess the cell surface expression of LLT1, 2B4, CS1, NKp30 and NKp46 on immune cells, PBMCs were isolated from the whole blood of 42 ALL subjects and 20 healthy subjects as a control. Flow cytometry was used to detect the cell surface expression (FL) indicating the percentage of cells positive for the target receptor on each sample as indicated in Figure 3. Increased cell surface expression of LLT1 (69% positive cells) was observed at diagnosis (1BD) as compared to 45% positive cells in healthy subjects (*p* = 0.0025) in CD14+ monocytes (Figure 3a). After induction chemotherapy (2BD), there was a decrease in the LLT1 expression (52.1%), which was comparable to the expression in healthy subjects. CS1 expression was more than double at diagnosis (37%) than in healthy subjects (14.7%) (*p* = 0.0027) in CD14+ monocytes (Figure 3c). NKp46 cell surface expression on monocytes was also significantly elevated in ALL subjects (17%) as compared to only 3% in healthy subjects (*p* = 0.0160) at diagnosis (Figure 3d). NKp30 cell surface expression on monocytes (Figure 3e) was also elevated but was not statistically significant (*p* > 0.05), whereas 2B4 expression was downregulated at diagnosis and at the end of induction chemotherapy (*p* > 0.05) as compared to healthy subjects (Figure 3b).

### 2.4. Downregulation of Cell Surface Expression of Immune Receptors on T Cells (CD3+) of ALL Subjects after Treatment

An overall decrease in the expression of receptors was observed on CD3+ T cells at diagnosis (day 0) and post-induction chemotherapy (day 29) as compared to healthy subjects. LLT1 expression in CD3+ T cells of ALL subjects at diagnosis (day 0) showed a median fluorescence intensity ratio (MFIR) value of 2.24 as compared to 2.74 in healthy individuals, which further decreased to 1.68 after induction chemotherapy treatment (*p* = 0.0247), as shown in Figure 4a.

A similar trend was observed in 2B4 (MFIR = 2.07), CS1 (MFIR = 2.39) and NKp46 (MFIR = 2.04) expression on T cells of ALL subjects after induction chemotherapy (day 29) treatment, where expression demonstrated a decrease upon comparison to that of healthy donors of 3.48 (*p* = 0.0171), 3.89 (*p* = 0.0111) and 3.16 (*p* = 0.0473), respectively (Figure 4b–d). The expression of NKp30 pre- and post-treatment did not show any significant (*p* > 0.05) change (Figure 4e).

### 2.5. Cell Surface Expression of Immune Receptors in CD56+ NK Cells and CD19+ B Cells in ALL Subjects before and after Induction Chemotherapy

LLT1 cell surface expression was significantly higher in CD56+ NK cells isolated from the blood of ALL subjects at diagnosis and at the end of induction chemotherapy (34.50% and 36.18%, respectively) as compared to the expression in healthy (12.64%) subjects (*p* = 0.0035 and 0.0016, respectively; Figure 5).

The expression of NKp30 (56.57%) and NKp46 (54.2%) on NK cells was elevated in ALL subjects at diagnosis as compared to healthy subjects (46.1% and 49.9%, respectively), which further reduced after induction chemotherapy (45.1% and 43.9%, respectively; Figure 5d,e). There was no significant change in the expression of CS1 on NK cells in ALL subjects at diagnosis and at the end of induction chemotherapy as compared to healthy subjects (Figure 5c), whereas 2B4 expression was downregulated in ALL subjects (Figure 5b). No significant change was observed in the expression of the receptors in CD19+ B cells of ALL subjects at diagnosis and at the end of induction chemotherapy as compared to healthy subjects (Figure 5f–j).

### 2.6. Demographic Data of ALL Subjects

A total of 46 ALL subjects and 20 healthy subjects were recruited in this study. Four subjects were excluded from the analysis as they did not fulfill the inclusion criteria or failed to complete the study. Out of the 42 eligible subjects, 39 of them had the B-cell subtype of ALL, while only 3 had T-cell ALL, as shown in Figure 6 and Table 1. The subjects were categorizd into three age goups: 2–5, 6–14 and 15–20 years. In both the first and the second groups, there were 18 subjects (43%) each, while the third group had only 6 subjects (14%). The gender was equally distributed: 50% (*n* = 21) of subjects were female and 50% (*n* = 21) were males.

In addition, we also collected some clinical data to understand how the disease affected their cellular proliferation and cytogenetics. Our data showed a massive elevation in white blood cells (WBC) count, where 47% of the ALL subjects’ blood revealed WBC count more than 20,000 cells/µL and four of those subjects surpassed 100,000 cells/µL. On the other hand, 30% of the subjects had a low count below 5000 cells/µL, while 23% of the subjects had a WBC count within the normal range between 5000 and 10,000 cells/µL. Of the 42 subjects, 26% indicated the presence of hyperploidy, some of which are known as trisomies of chromosomes 4, 10 or 17, either in one, two or in all of them. A couple of the subjects had the high-risk B-ALL intrachromosomal amplification of chromosome 21 (iAMP21), while another couple of subjects were Philadelphia chromosome (Ph)/BCR/ABL-positive. Upon completion of induction chemotherapy, the minimal residual disease (MRD) of the majority of subjects (81%) came back negative, while in 14%, it remained positive, as shown in Figure 6 and Table 1. For 5% of the subjects, MRD was unknown.

## 3. Discussion

Relapse after treatment in ALL remains the greatest impediment to overall survival. This is due to the fact that relapse lowers the chances of overall survival (OS) from more than 90% after primary treatment to 50% or less after first relapse, which is reduced even more with subsequent relapses [40,41,42]. It has been shown that certain risk factors accompanying relapse can predict treatment efficiency and contribute to the anticipated prognosis. These risk factors can be time (the faster the relapse the lower the chances of treatment effectiveness), immunophenotype (T-cell relapses tend to be more aggressive than B-cell malignancies and put the patient by default in the high-risk group regardless of the timing) or the site of relapse (bone marrow relapses are usually worse than extramedullary) [14,43,44]. 

The demographic data indicate the usual trend of most of ALL subjects being B-ALL rather than T-ALL. One of the T-ALL subjects’ disease was very aggressive and the subject did not survive the completion of the induction chemotherapy. T-cell ALL usually tends to be more invasive and resistant to treatment than B-cell ALL [45,46].

Despite the diverse repertoire of killing strategies utilized by NK cells, cancer cells often avoid activating NK cells and spread to other locations—causing disease progression by direct and indirect mechanisms [47]. The RNA-seq data analysis from the St. Jude PeCan data portal and our cohort study mRNA expression data (Figure 1 and Figure 2) showed overexpression of LLT1 in the ALL subject samples. The flow cytometry data confirmed that there was a significant increase in the surface expression of LLT1 on both monocytes and NK cells of ALL subjects. Previously, we have shown that prostate cancer cells and triple-negative breast cancer (TNBC) cells overexpress LLT1 as a mechanism to evade NK-cell immunosurveillance [32,33,48]. Studies have shown that LLT1 can inhibit the ability of NK cells to target glioma, prostate cancer, squamous cell carcinoma, B-cell lymphomas, lung cancer and triple-negative breast cancer cells. LLT1 on tumor cells interact with an NK-cell inhibitory receptor, NKRP1A (CD161), leading to the inhibition of NK-cell-mediated cytotoxicity of tumor cells [32,33,35,49,50,51,52,53]. Targeting LLT1–NKRP1A interaction provides an attractive alternative for overcoming tumor escape mechanism in different cancers.

In contrast to the inhibitory effect of LLT1–NKRP1A interaction on NK cells, CD161 (NKRP1A) expression on T cells causes T-cell proliferation upon interaction with LLT1. The interaction of LLT1 with CD161 may, on the one hand, inhibit NK-cell effector functions and, on the other hand, co-stimulate T cells. This would suggest that LLT1/CD161 interaction participates in the sequential involvement of NK cells and later T cells in the initiation of adaptive immune responses, with the LLT1/CD161 interaction shutting down NK-cell activation while co-stimulating T cells [52,53,54]. LLT1 expression on NK cells has a synergistic effect on their cytokine release behavior, indicating the chronic activation of NK cells, which may be an indicator for NK cells’ exhaustion and their inability to recognize malignant cells during or briefly after treatment, and they would no longer carry out their cytotoxic effects even upon binding to activating receptors, such as CS1, 2B4 NKp30 and NKp46, which would drive the surrounding blood cells to elevate their expression of these receptors. It may also promote the possibility of relapses since NK cells’ immunosurveillance is compromised [26,55,56].

The signaling lymphocyte-activating molecule family (SLAM), which 2B4 (SLAMF4) and CS1 (SLAMF7) belong to, also plays an important role in NK-cell regulation. 2B4 and CS1 are naturally expressed on monocytes and T cells. Depending on the context and the presence of aiding proteins, they can act as either activating or inhibitory receptors. CS1 is a homophilic receptor that is self-activating, while 2B4 has a high affinity to CD48 as its ligand. CS1 and LLT1 were observed to be overexpressed in monocytes (Figure 3). Overexpression of CS1 in multiple myeloma (MM) led to the approval of Elotuzumab as the first FDA-approved monoclonal antibody-targeting CS1 for the treatment of MM [57]. Our previous studies have shown that CS1 is expressed on activated monocytes and plays an inhibitory role by reducing the production of proinflammatory cytokines by LPS-activated monocytes [58]. Studies have revealed SLAMF7 to be a potent inhibitor of the monocyte-derived proinflammatory chemokine CXCL10 (IP-10) and other CXCR3 ligands, except in a subset of HIV+ patients termed SLAMF7 silent (SF7S) [59]. Moreover, SLAMF7^high^CD16− monocytes and IL-1ra were correlated with myelofibrosis (MF) onset in myeloproliferative neoplasm patients who harbor JAK2V617F. Elotuzumab (anti-CS1 mAb) suppressed fibrocyte differentiation and MF progression in vitro and in vivo [60]. Overexpression of CS1 on monocytes in ALL subjects may have similar inhibitory effects. In pathological contexts, LLT1 has been reported to be expressed on monocytes of synovial fluid and macrophages within synovial tissues of patients with rheumatoid arthritis [61]. The molecular mechanisms behind the regulation of LLT1 in several cell types is still poorly understood. 2B4 expression was downregulated in monocytes and T cells (Figure 3b and Figure 4b). Previous studies have shown that the lack of or dysfunction of 2B4 has been linked to immunodeficiency as well as its involvement as a costimulatory molecule to CD8+ T cells [35,36,62].

Even though NKp30 and NKp46 are known to be expressed on NK cells’ surface as part of the natural cytotoxicity-activating receptors (NCRs), they also have been shown to be expressed on some T-cell subsets [37]. High expression of NKp46 and NKp30 in the CD56+NK cells of ALL subjects was in contrast to previously reported studies. NK cells from T-ALL patients had reduced expression of the activating receptors NKp46 and DNAM-1, but not NKG2D [63]. Studies have also shown that in some cancers, NKp30 and NKp46 expression is altered or downregulated, which possibly creates an immunocompromised tumor microenvironment, which in turn helps in promoting tumor progression [64]. Interestingly, we did not observe any appreciable changes in the receptor expression in the CD19+ B-cell subsets of ALL subjects both at diagnosis and at the end of induction chemotherapy as compared to healthy subjects.

In conclusion, we have shown that LLT1 was overexpressed in PBMCs at the transcript level and at the cell surface in monocytes, T cells and NK cells. Increased expression of CS1 and NKp46 was observed on monocytes of ALL subjects at diagnosis. A decrease of LLT1, 2B4, CS1 and NKp46 on T cells of ALL subjects was also observed post-induction chemotherapy. The results indicate that the differential expression of the receptors/ligand may play a role in the T-cell- and NK-cell-mediated immune surveillance of pediatric ALL. Since 2B4 and NKp46 are already being used as transmembrane domains for CAR-NK structuring, and CS1 and 2B4 have shown promising results in CAR-T and CAR-NK immunotherapy, further studies are warranted to assess the functional role of these receptors, especially LLT1 in a larger cohort of ALL subjects to develop an immunotherapeutic alternative for childhood ALL.

## 4. Materials and Methods

### 4.1. RNA Sequencing Data Collection

RNA sequencing data were collected from the St. Jude Children’s Research Hospital Pediatric Cancer Genome project (PeCan data portal: https://pecan.stjude.cloud; accessed on 24 June 2022) and analyzed for expression of receptors: 2B4, CS1, LLT1, NKp30 and NKp46 in B-ALL and T-ALL subject samples.

### 4.2. Subjects and Healthy Volunteers

Newly diagnosed ALL subjects, aged between 2 and 21 years old, were enrolled in the study at the Hematology and Oncology Clinic at Cook Children’s Medical Center (CCMC), Fort Worth, TX, with informed consent/assent obtained by Dr. Paul Bowman, MD and nursing staff as per IRB approval from UNTHSC and CCMC (UNTHSC IRB# 2008-094 & CCMC IRB# 2008-57). Additionally, healthy subjects under the age of 21, who attend regular medical visits at the pediatric clinic at UNTHSC Health Pavilion, were enrolled. 

### 4.3. Blood Collection

After consent was obtained, the staff collected one blood sample (8 mLs) from the patient before any treatment. This is referred to as the first blood draw (1BD). Another blood sample was collected 29 days later, after the initial chemotherapy, which typically lasts 28 days. This is referred to as the second blood draw (2BD). A total of 42 ALL subjects were enrolled. Only one blood draw was collected from healthy subjects, and 20 healthy subjects were enrolled.

### 4.4. PBMC Isolation

Peripheral blood mononuclear cells (PBMCs) were isolated from ethylenediamine tetra acetic acid (EDTA)-treated whole-blood samples by Histopaque-1077 (Sigma Chemicals, St Louis, MO, USA) density gradient centrifugation using LeucoSep tubes (Greiner, Monroe, NC, USA). The remaining red blood cells were lysed with ACK lysis buffer. The PBMCs were then separated into two parts: one part was used for flow cytometric analysis of immune receptor expression and the second part was used to isolate mRNA for qRT-PCR analysis.

### 4.5. qRT-PCR Analysis

Five million cells were dissolved with 1 ml of RNA STAT-60. RNA was extracted by chloroform and precipitated by isopropanol. After resuspension with 0.1% diethylpyrocarbonate (DEPC) water, RNA purity and concentration was determined by measuring optical density. Then, 2 μg of RNA was used for cDNA synthesis in the presence of random primer mix (NEB). After RT reaction, 100 ng of cDNA was used as a template, and Taqman mastermix and Taqman primers for 2B4, CS1, LLT1, NKp30 and NKp46 were used on the Eppendorf Realplex2 to perform the PCR reaction. Fold change of expression between healthy subjects and patient subjects was calculated using the ΔΔCT method, which compares the CT value, defined as the number of cycles required for the fluorescence to reach threshold, of the target gene and the housekeeping gene (GAPDH) of one sample group to a control sample group. The efficiency of PCR was 90–100%. The results presented are an average of three independent experiments.

### 4.6. Flow Cytometry

A Beckman Coulter FC500 was used, and the populations of immune cells were gated by forward and side scatter to separate the lymphocytes and monocytes from other cells by size and granularity. Various fluorochrome markers were used to differentiate the immune cells: FITC-conjugated anti-human CD3 monoclonal antibody (mAb) for T cells, PE-Texas red-conjugated anti-human CD19 mAb for B cells, APC-conjugated anti-human CD56 mAb for NK cells, and APC-Cy7-conjugated CD14 mAb for monocytes. Each immune receptor of interest and ligands were labeled with PE. Samples were stained with anti-2B4, anti-CS1, anti LLT1, anti-NKp30 and anti-NKp46 mAb to determine percentage and quantitative cell surface expression of these receptors. All the antibodies used were from Biolegend, San Diego, CA, USA.

### 4.7. Statistical Analysis

To analyze the differences in immune receptor expression between healthy subjects, ALL subject samples at diagnosis and at the end of induction chemotherapy, a multivariate analysis of variance (MANOVA) test was performed in SAS version 9.4. MANOVA allows researchers to account for correlation between multiple measurements obtained from the same subject. In the presence of statistically significant finding for overall difference using MANOVA, we evaluated each item individually using the ANOVA approach. Significance was determined at alpha = 0.05.

## Figures and Tables

**Figure 1 ijms-24-03860-f001:**
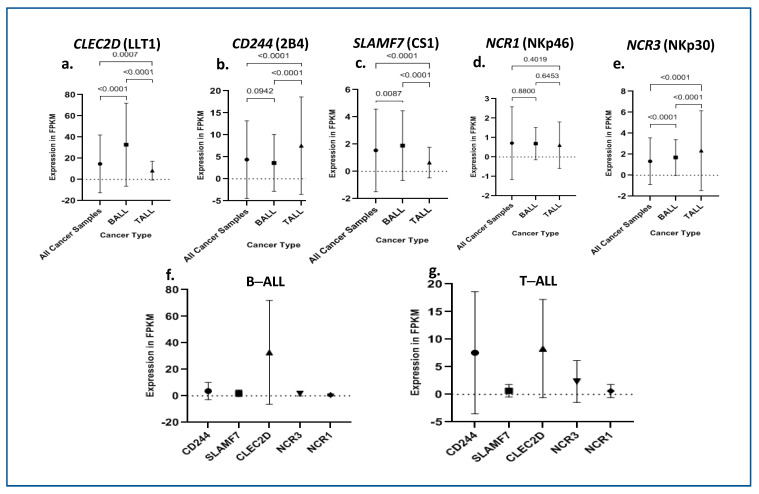
RNA–seq expression in ALL subtypes and other pediatric cancers. (**a**–**e**) Expression of each of the receptors in B–ALL and T–ALL subjects in comparison with all other pediatric cancers. (**f**) Comparison of expression of LLT1, 2B4, CS1, NKp46 and NKp30 in B–ALL subjects. (**g**) Comparison of expression of LLT1, 2B4, CS1, NKp46 and NKp30 in in T–ALL subjects.

**Figure 2 ijms-24-03860-f002:**
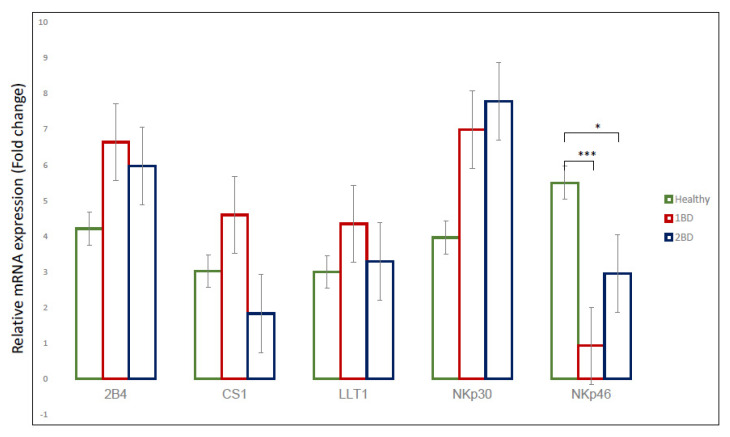
mRNA expression of *2B4, CS1, LLT1, NKp30 and NKp46* in ALL subjects**.** qRT–PCR data indicating mRNA expression of immune receptors from PBMC isolated from ALL subjects’ blood pre– and post–induction chemotherapy compared to healthy subjects. 1BD, first blood draw indicates sample obtained at diagnosis (day 0). 2BD, second blood draw indicates sample obtained at the end of induction chemotherapy (day 29). * *p* < 0.05, *** *p* < 0.005.

**Figure 3 ijms-24-03860-f003:**
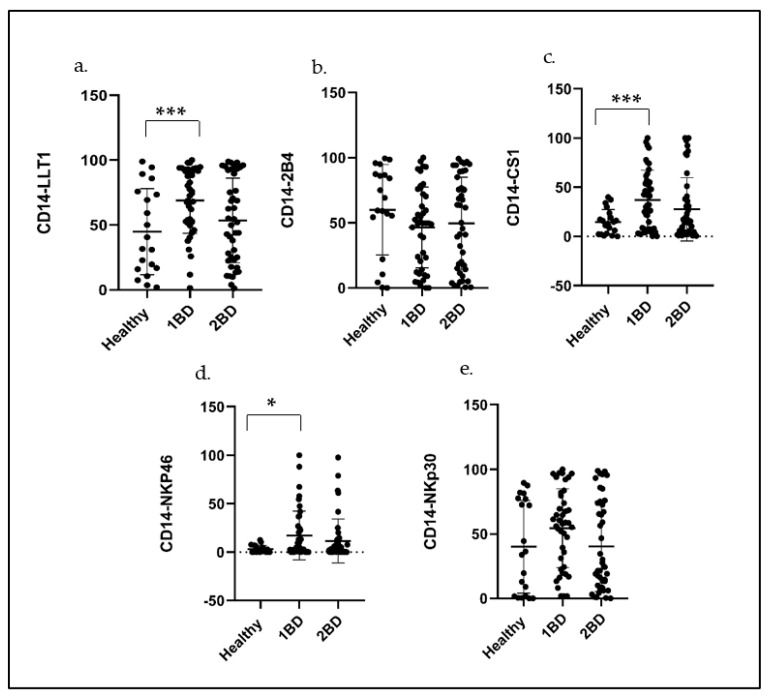
Expression of LLT1, 2B4, CS1, NKp30 and NKp46 on monocytes (CD14+) of ALL subjects compared to healthy subjects. (**a**–**e**) Flow cytometry analysis exhibit percentage of positive cells for the indicated receptors in CD14+ monocytes in ALL subjects and healthy subjects at diagnosis and post–induction chemotherapy. 1BD, first blood draw indicates sample obtained at diagnosis (day 0). 2BD, second blood draw indicates sample obtained at the end of induction chemotherapy (day 29). * *p* < 0.05, *** *p* < 0.005.

**Figure 4 ijms-24-03860-f004:**
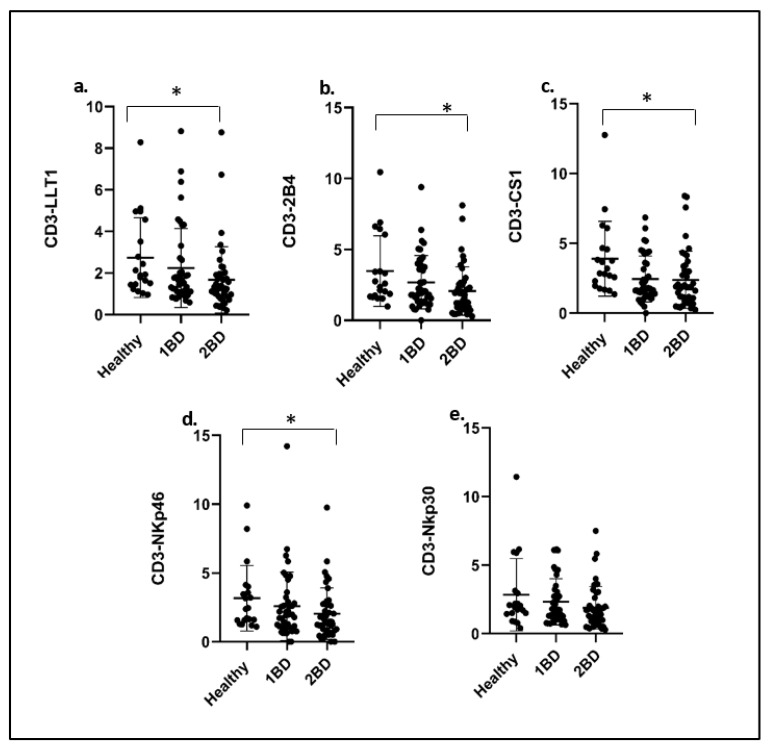
Expression of LLT1, 2B4, CS1, NKp30 and NKp46 expression on T cells (CD3+) of ALL subjects. (**a**–**e**) Flow cytometry analysis exhibits the cell surface expression of indicated receptors on T cells of ALL subjects compared to healthy subjects indicated by the median fluorescence intensity ratio (MFIR). 1BD, first blood draw indicates sample obtained at diagnosis (day 0). 2BD, second blood draw indicates sample obtained at the end of induction chemotherapy (day 29). * *p* < 0.05.

**Figure 5 ijms-24-03860-f005:**
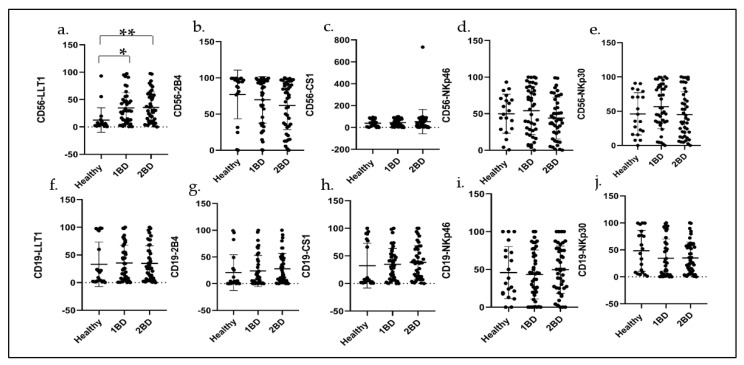
Expression of LLT1, 2B4, CS1, NKp30 and NKp46 on NK and B cells of ALL subjects before and after induction chemotherapy. Flow cytometry analysis exhibit cell surface expression of indicated receptors on (**a**–**e**) NK cells and (**f**–**j**) CD19+ B cells of ALL subjects compared to healthy subjects indicated by percentage of positive cells (FL) for the indicated receptors. 1BD, first blood draw indicates sample obtained at diagnosis (day 0). 2BD, second blood draw indicates sample obtained at the end of induction chemotherapy (day 29). * *p* < 0.05, ** *p* < 0.01.

**Figure 6 ijms-24-03860-f006:**
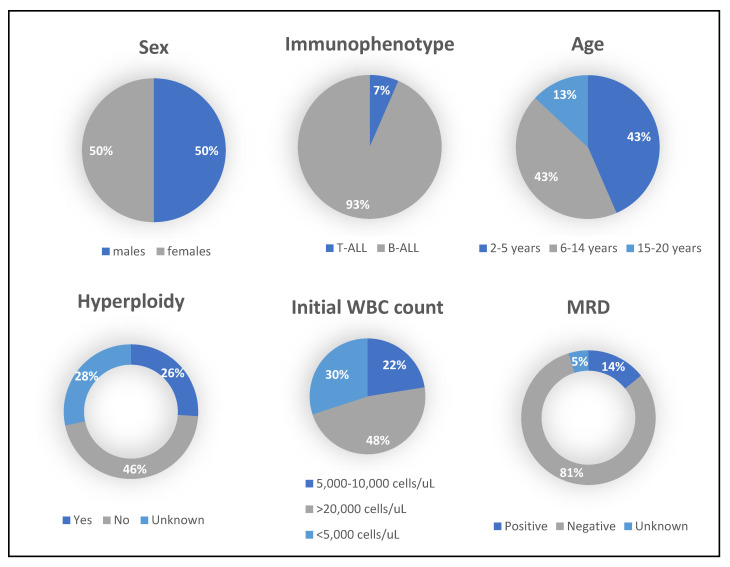
Demographics of ALL subjects**.** WBC: white blood cells. MRD: minimal residual disease.

**Table 1 ijms-24-03860-t001:** Demographic and clinicopathological features of ALL subjects.

Clinical Features	No. of Patients
Sex	
Male	21
Female	21
ALL Subtype	
B-ALL	39
T-ALL	3
Age	
2–5	18
6–14	18
15–20	6
MRD	
Positive	6
Negative	34
Unknown	2
Initial WBC count (Cells/ul)	
<5000	12
5000–10,000	9
>20,000	19
Hyperdiploidy	
Yes	12
No	19
Unknown	11

## Data Availability

The RNA sequencing data presented in this study were obtained from the publicly available St. Jude Children’s Research Hospital Pediatric Cancer Genome database (PeCan data portal: https://pecan.stjude.cloud; accessed on 24 June 2022).

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
