# Peer review of "Differential Expression of LLT1, SLAM Receptors CS1 and 2B4 and NCR Receptors NKp46 and NKp30 in Pediatric Acute Lymphoblastic Leukemia (ALL)"

_ijms, 2023, doi:10.3390/ijms24043860_

Round 1

Reviewer 1 Report

In this manuscript, Powers et al represented the differential expressions of a few genes in pediatric Acute Lymphoblastic Leukemia (ALL). First, they analyzed the expression of genes using the available data from a database. Then they recruited patients and analyzed mRNA expression in whole blood. Those expression data were validated in a different types of immune cells using FACS analysis. However, there are a few major comments, which need to be resolved before considering the manuscript for publication in this Journal.

a) Author analyzed protein expression in different types of cells. It would be great if they could analyze mRNA expression in the same type of cells instead of PBMC at least in 5 normal and 5 patient subjects.

 b) In the introduction (last papa) describes the source and method of sample collection. Please relocate the para in the method section.

c) In RNA seq data, generally several hundreds of genes are identified as deregulated genes. From the list of those genes, how authors prioritized a few selected genes? Please clarified it in the result and method section.

d) Authors only identified the deregulation of 5 genes. What is the mechanism of upregulation or downregulation in ALL? Please see some mechanistic aspects. Is it genetic or epigenetic regulation?

e) Authors check the expression at the mRNA level. Please validate your mRNA data at the protein level in at least 5 control and 5 ALL samples by western blot.

f) Please provide patients' demographic data as a table for both RNA seq as well as newly collected samples showing the epidemiological and clinicopathological features.

g) In recent discoveries, Stress is the most important factor, which could induce tumorigenesis and promote cancer development via modulation of genetic and epigenetic changes. Stress has the potential power to generate almost all the hallmarks of cancer in a healthy person. Moreover, it could modulate the chemotherapeutic efficacy via the regulation of differential gene expressions in different types of cancer (https://doi.org/10.3389/fonc.2020.01492;  https://doi.org/10.1038/s41568-021-00395-5). In this regard, it will be more logical to see the expression of those genes in different types of immune cells and stratified by the stress level of the patients before and after chemotherapy.

Author Response

In this manuscript, Powers et al represented the differential expressions of a few genes in pediatric Acute Lymphoblastic Leukemia (ALL). First, they analyzed the expression of genes using the available data from a database. Then they recruited patients and analyzed mRNA expression in whole blood. Those expression data were validated in a different types of immune cells using FACS analysis. However, there are a few major comments, which need to be resolved before considering the manuscript for publication in this Journal.

  1. Author analyzed protein expression in different types of cells. It would be great if they could analyze mRNA expression in the same type of cells instead of PBMC at least in 5 normal and 5 patient subjects.

Response: We highly appreciate the reviewer’s comment and ideally would have analyzed the mRNA expression in different cell types instead of PBMC from the ALL patients. But one of our biggest limitation was the sample volume which is a great challenge when you work with pediatric patients. Our day 0 sample collection would take place simultaneously with all other blood draws for diagnostic purposes to identify whether the patient has leukemia or not. At that moment the child and the family have a lot of emotional, psychological and physical stress. We would consider ourselves lucky if we got 5 mls of blood from the patient. Considering all of those limitations we just could not isolate individual cell types and run mRNA qRT-PCR analysis on them. Therefore, we decided to do flow cytometry which would provide valuable information about the cell surface expression of the receptors which is very important for the functional outcome of the receptor-ligand interactions especially in the context of NK cell function.

2.  In the introduction (last papa) describes the source and method of sample collection. Please relocate the para in the method section.

Response: We have removed the last paragraph from the introduction where the source and method of collection was described. The methods section already has all of the details of sample collection.

3. In RNA seq data, generally several hundreds of genes are identified as deregulated genes. From the list of those genes, how authors prioritized a few selected genes? Please clarified it in the result and method section.

Response: It is true that RNAseq data can identify hundreds of genes that are either upregulated or downregulated but for the purpose of this study we queried the major activating and inhibitory receptor/ligands in the St. Jude Pediatric Cancer genomic (PeCan) database. As mentioned in the introduction section, three of the five receptors that were selected i.e. 2B4, CS1 and LLT1 were cloned, identified and characterized in our laboratory and so we were interested to further examine the relevance of these receptors in a pediatric ALL population and compare their expression with the NCR receptors: NKp46 and NKp30 that has been widely studied in ALL research. Therefore, these 5 receptors/ligand were chosen for this study.

4. Authors only identified the deregulation of 5 genes. What is the mechanism of upregulation or downregulation in ALL? Please see some mechanistic aspects. Is it genetic or epigenetic regulation?

Response: This is an excellent question and we thank the reviewer for pointing that out. Specifically, in ALL we do not know the exact mechanism of either upregulation or down regulation of these receptors but the possibility of either genetic or epigentic regulation cannot be ruled out. Further mechanistic studies are warranted to ascertain the exact mechanism of action which is beyond the scope of this paper but would be a plausible next step for this project.

5. Authors check the expression at the mRNA level. Please validate your mRNA data at the protein level in at least 5 control and 5 ALL samples by western blot.

Response: We appreciate the reviewer’s comment about validating the mRNA expression data at the protein level in at least 5 control and 5 ALL samples by western blot. However, we want to point out that we did confirm cell surface protein expression of the receptors in T cells, B cells, NK cells and monocytes by conducting flow cytometry on all 42 patients and 20 control subjects and the data has been reported in Figures 3 – 5. Flow cytometry helps in determining cell surface protein expression as opposed to the total protein content determined by western blot unless you do cell fractionation to determine membrane, cytosolic and nuclear fractions. In the context of NK cells and T cells, the receptor-ligand interactions are important and therefore we focused on determining the cell surface protein expression by flow cytometry instead of western blot. Since, our IRB protocol does not permit us to re-contact the patients it is not possible to get any more blood samples from them to conduct any additional western blot analysis.

6. Please provide patients' demographic data as a table for both RNA seq as well as newly collected samples showing the epidemiological and clinicopathological features.

Response: The RNAseq expression data was obtained from the St. Jude Pediatric Cancer genomic (PeCan) database where a total of 729 B-ALL subjects and 313 T-ALL subjects’ data was analyzed. It is practically impossible to show the patient demographic, epidemiological and clinicopathological features in a table for that large cohort. Besides that, the patient demographic, epidemiological and clinicopathological is not available on St. Jude’s PeCan portal. However, we have added a table to show the demographic, epidemiological and clinicopathological features for the 42 patients that were recruited in our cohort study.

7. In recent discoveries, Stress is the most important factor, which could induce tumorigenesis and promote cancer development via modulation of genetic and epigenetic changes. Stress has the potential power to generate almost all the hallmarks of cancer in a healthy person. Moreover, it could modulate the chemotherapeutic efficacy via the regulation of differential gene expressions in different types of cancer (https://doi.org/10.3389/fonc.2020.01492;  https://doi.org/10.1038/s41568-021-00395-5). In this regard, it will be more logical to see the expression of those genes in different types of immune cells and stratified by the stress level of the patients before and after chemotherapy.

 Response: We completely agree with the reviewer’s comments about stress being an important factor for inducing tumorigenesis and promoting cancer development via modulation of genetic and epigenetic changes. In the context of leukemia, there have been studies that have reported occupational hazards and exposure to organic solvents like benzene to cause leukemia. Further studies are warranted to examine the role of these stressors in modulating the expression of immune receptors and their role in the induction and progression of leukemia in a pediatric population.

Reviewer 2 Report

Research article “Differential expression of LLT1, SLAM receptors CS1 and 2B4 and NCR receptors NKp46 and NKp30 in pediatric Acute Lymphoblastic Leukemia (ALL)” by Powers et. al. provides a broad view of the expression profiles of immune receptors in PBMCs of B-ALL and T-ALL subjects by single-cell RNA sequencing. Authors report in ALL subjects, alterations in expression profile of receptors pre- and post-induction chemotherapy were observed.

The results of this clinically relevant study on pediatric ALL cases show the differential expression of receptors/ligand that may play role in T cell and NK-cell mediated immune surveillance. It’s interesting to observe that induction chemotherapy led to decrease in the LLT1 expression. At diagnosis stage, a more than double difference in CS1 expression compared to healthy subjects is also clinically relevant to understand disease presentation and treatment response. The material and methods section provides all necessary details related to sample collection, processing, qRT PCR and statistical analysis.

The research article can be accepted in its current form. However, further studies on larger cohort of ALL subjects will be beneficial to assess the translational relevance of LLT1 expression in terms of development as a target of immuno-therapeutic relevance for childhood ALL.

Author Response

Research article “Differential expression of LLT1, SLAM receptors CS1 and 2B4 and NCR receptors NKp46 and NKp30 in pediatric Acute Lymphoblastic Leukemia (ALL)” by Powers et. al. provides a broad view of the expression profiles of immune receptors in PBMCs of B-ALL and T-ALL subjects by single-cell RNA sequencing. Authors report in ALL subjects, alterations in expression profile of receptors pre- and post-induction chemotherapy were observed.

The results of this clinically relevant study on pediatric ALL cases show the differential expression of receptors/ligand that may play role in T cell and NK-cell mediated immune surveillance. It’s interesting to observe that induction chemotherapy led to decrease in the LLT1 expression. At diagnosis stage, a more than double difference in CS1 expression compared to healthy subjects is also clinically relevant to understand disease presentation and treatment response. The material and methods section provides all necessary details related to sample collection, processing, qRT PCR and statistical analysis.

The research article can be accepted in its current form. However, further studies on larger cohort of ALL subjects will be beneficial to assess the translational relevance of LLT1 expression in terms of development as a target of immuno-therapeutic relevance for childhood ALL.

Response: We greatly appreciate the reviewers’ comments for accepting the manuscript in its current form. We do agree that further studies in a larger cohort of ALL subjects will be beneficial to assess the translational relevance of LLT1 expression in terms of development as a target of immuno-therapeutic relevance for childhood ALL.

Reviewer 3 Report

Dear Authors, 

please, find my comments on the manuscript below:

1. Double notation of the analysed receptors should be avoided. I suggest using one abbreviation for a particular receptor instead of both, for example, "LLT1" instead of "LLT1 (CLEC2D)"  or "CLEC2D (LLT1)". Now, there is a different notation for the same receptor in both the text of the manuscript as well as in the figures, which causes the text to be less readable.

2. Titles of subsections of the manuscript should be reconsidered and reworded. Now, they present some conclusions but do not correspond well to the contents of the subsections. For example, 2.1. subsection presents data about RNA-seq expression for five analysed receptors and not only for the LLT1 expression. The same comment is valid for titles of figures 2-5. I recommend rewriting them so that they inform about the content of the figures and not only present selected conclusions. 

3. Ad subsection 2.1. : the description of the results of the analysis of the expression profile of the immune receptors is not full. You mentioned about CLEC2D, CD244, and NCR3 but what about SLAMF7 and NKp46? All data presented in Figure 1 should be commented on in this subsection. 

4. Ad 2.2 subsection: It should be clearly stated which increments/decrements in expression are statistically significant and which are not. In Figure 2 there were statistically significant differences only for the NKp46 expression presented. 

5. Ad 2.3. subsection: You mentioned the expression of LLT1 was increased in 1BD in comparison to healthy control. Figure 3 should be indicated which difference in LLT1 expression is statistically significant. Now, three asterisks are placed centrally above the data points, so it could suggest that the significant difference is between the three analysed cohorts (healthy, 1BD, 2 BD) globally as a result of ANOVA. I should be clarified. Additionally, you mentioned LLT1 expression decreased after induction chemotherapy. But in comparison to what? 1BD. P-value should be provided in Figure 3a. 

6. Ad 2.4. subsection: Data regarding NKp30 presented in Figure 4e should be described. P-value should be provided in Figure 3. 

7. Ad 2.6. subsection: Demographic data should be calculated for 42 patients enrolled and not excluded further from analysis, because expression analysis was performed in 42 subjects and not 46.

Line 209: 84% instead of 86% was given for MDR negative cases.

8. Ad 4.5 subsection: The housekeeping gene used in the analysis should be reported. Moreover, efficiencies of qPCR reaction for all primer pairs used should be measured and reported. These efficiencies are critical parameters in relative expression level calculation. Was there a correction for qPCR efficiency used in deltadelta CT method? 

9. Ad 4.7 subsection: Why MANOVA was chosen for analysis? Was the conformity of data with normal distribution checked? How? 

 Best regards

Author Response

  1. Double notation of the analyzed receptors should be avoided. I suggest using one abbreviation for a particular receptor instead of both, for example, "LLT1" instead of "LLT1 (CLEC2D)"  or "CLEC2D (LLT1)". Now, there is a different notation for the same receptor in both the text of the manuscript as well as in the figures, which causes the text to be less readable.

Response: We appreciate the reviewers comments about the double notations and at times it would become confusing but we want to state that LLT1 is the name of the receptor and its protein. However, the gene for LLT1 is CLEC2D. Similarly, as per NCBI the genes for 2B4, CS1, NKp46 and NKp30 are CD244, SLAMF7, NCR1 and NCR3 respectively. Therefore, the mRNA/gene data showed in Figures 1 and 2 have the gene notation of CLEC2D, CD244, SLAMF7, NCR1 and NCR3. We have also provided the names of the receptor along with it to correctly identify it. We have revised the text and figures to correctly notate and differentiate the gene from its protein form.

  1. Titles of subsections of the manuscript should be reconsidered and reworded. Now, they present some conclusions but do not correspond well to the contents of the subsections. For example, 2.1. subsection presents data about RNA-seq expression for five analysed receptors and not only for the LLT1 expression. The same comment is valid for titles of figures 2-5. I recommend rewriting them so that they inform about the content of the figures and not only present selected conclusions. 

Response: As per the recommendation of the reviewer, subsection 2.1 and all subsequent titles of subsection have been updated to provide a clearer understanding of the results and the figures to the reader.

  1. Ad subsection 2.1. : the description of the results of the analysis of the expression profile of the immune receptors is not full. You mentioned about CLEC2D, CD244, and NCR3 but what about SLAMF7 and NKp46? All data presented in Figure 1 should be commented on in this subsection. 

Response: As recommended by the reviewer we have updated the text and included a complete analysis of the receptors including SLAMF7 and NKp46 as presented in Figure 1 in the description under subsection 2.1.

  1. Ad 2.2 subsection: It should be clearly stated which increments/decrements in expression are statistically significant and which are not. In Figure 2 there were statistically significant differences only for the NKp46 expression presented. 

Response: As recommended by the reviewer we have clearly stated the increments or decrements in expression that were statistically significant and provided the p values as well in subsection 2.2.

  1. Ad 2.3. subsection: You mentioned the expression of LLT1 was increased in 1BD in comparison to healthy control. Figure 3 should be indicated which difference in LLT1 expression is statistically significant. Now, three asterisks are placed centrally above the data points, so it could suggest that the significant difference is between the three analysed cohorts (healthy, 1BD, 2 BD) globally as a result of ANOVA. I should be clarified. Additionally, you mentioned LLT1 expression decreased after induction chemotherapy. But in comparison to what? 1BD. P-value should be provided in Figure 3a. 

Response: We admit that the results presented were a little confusing in subsection 2.3. As per the reviewer’s recommendation we have revised the figure to clearly show that statistically significant difference in LLT1 expression (3 stars) was indeed observed in 1 BD as compared to healthy subjects showing a p value of 0.0025. Additionally, we have also clarified the decrease in LLT1 expression in 2BD as compared to healthy subjects. The text in subsection 2.3 has been revised.

  1. Ad 2.4. subsection: Data regarding NKp30 presented in Figure 4e should be described. P-value should be provided in Figure 3. 

Response: Data regarding NKp30 expression has been now added in subsection 2.4. The expression of NKp30 pre and post treatment didn’t show any significant change (p>0.05). P-values have been added to the text.

  1. Ad 2.6. subsection: Demographic data should be calculated for 42 patients enrolled and not excluded further from analysis, because expression analysis was performed in 42 subjects and not 46.

Line 209: 84% instead of 86% was given for MDR negative cases.

Response: The demographic data has been updated with analysis for 42 patients in subsection 2.6. Figure 6 has also been updated. Figure 6 was also updated to include the unknown MRD of 2 samples. Therefore, cases with negative MRD were quantified to be 81% of all cases, positive cases were 14% and unknown cases were 5%. Line 209 has been revised.

  1. Ad 4.5 subsection: The housekeeping gene used in the analysis should be reported. Moreover, efficiencies of qPCR reaction for all primer pairs used should be measured and reported. These efficiencies are critical parameters in relative expression level calculation. Was there a correction for qPCR efficiency used in deltadelta CT method? 

Response: The housekeeping gene used was GAPDH and has been reported in the description of the methods in subsection 4.5. The efficiency of PCR was 90 – 100%. To obtain unbiased efficiency-corrected results, absolute quantification with a single undiluted calibrator with a known target concentration and efficiency values derived from the amplification curves of the calibrator and the patient samples were determined.

  1. Ad 4.7 subsection: Why MANOVA was chosen for analysis? Was the conformity of data with normal distribution checked? How? 

Response: MANOVA allows researchers to account for correlation between multiple measurements obtained from the same subject. In the presence of statistically significant finding for overall difference using MANOVA, we evaluated each item individually using the ANOVA approach. We created box-plots and histograms for individual items. We did not conduct a formal test for multivariate normality.  

The multiple items LLT1, 2B4, CS1, NKp30 and NKp46 were collected from the same subject. Hence, they are correlated. As MANOVA assume multiple outcomes are correlated we used the more conservative approach MANOVA to test them jointly rather than testing them individually. For correlated items MANOVA is considered more rigorous rather than using ANOVA and testing each item individually. We have revised the text in subsection 4.7 to provide more clarity about the statistical analysis.

Round 2

Reviewer 1 Report

1.      Author analyzed protein expression in different types of cells. It would be great if they could analyze mRNA expression in the same type of cells instead of PBMC at least in 5 normal and 5 patient subjects.

Response: We highly appreciate the reviewer’s comment and ideally would have analyzed the mRNA expression in different cell types instead of PBMC from the ALL patients. But one of our biggest limitation was the sample volume which is a great challenge when you work with pediatric patients. Our day 0 sample collection would take place simultaneously with all other blood draws for diagnostic purposes to identify whether the patient has leukemia or not. At that moment the child and the family have a lot of emotional, psychological and physical stress. We would consider ourselves lucky if we got 5 mls of blood from the patient. Considering all of those limitations we just could not isolate individual cell types and run mRNA qRT-PCR analysis on them. Therefore, we decided to do flow cytometry which would provide valuable information about the cell surface expression of the receptors which is very important for the functional outcome of the receptor-ligand interactions especially in the context of NK cell function.

Reviewer’s Comment: I agree with the Authors regarding primary sample. I think, the alternative of primary samples is cell lines. Please check it in different ALL and other cell lines/ EVB transformed cells. Authors also have the scope to check the expression in different cell derived from same iPSC (if possible).

2. In the introduction (last papa) describes the source and method of sample collection. Please relocate the para in the method section.

Response: We have removed the last paragraph from the introduction where the source and method of collection was described. The methods section already has all of the details of sample collection.

Reviewer’s Comment: I agree. Thank you.

3. In RNA seq data, generally several hundreds of genes are identified as deregulated genes. From the list of those genes, how authors prioritized a few selected genes? Please clarified it in the result and method section.

Response: It is true that RNAseq data can identify hundreds of genes that are either upregulated or downregulated but for the purpose of this study we queried the major activating and inhibitory receptor/ligands in the St. Jude Pediatric Cancer genomic (PeCan) database. As mentioned in the introduction section, three of the five receptors that were selected i.e. 2B4, CS1 and LLT1 were cloned, identified and characterized in our laboratory and so we were interested to further examine the relevance of these receptors in a pediatric ALL population and compare their expression with the NCR receptors: NKp46 and NKp30 that has been widely studied in ALL research. Therefore, these 5 receptors/ligand were chosen for this study.

Reviewer’s Comment:  I agree.

4. Authors only identified the deregulation of 5 genes. What is the mechanism of upregulation or downregulation in ALL? Please see some mechanistic aspects. Is it genetic or epigenetic regulation?

Response: This is an excellent question and we thank the reviewer for pointing that out. Specifically, in ALL we do not know the exact mechanism of either upregulation or down regulation of these receptors but the possibility of either genetic or epigentic regulation cannot be ruled out. Further mechanistic studies are warranted to ascertain the exact mechanism of action which is beyond the scope of this paper but would be a plausible next step for this project.

Reviewer’s Comment: If so, author could show some mechanistic functional aspects of upregulation of the said genes in ALL cell line (s).

5. Authors check the expression at the mRNA level. Please validate your mRNA data at the protein level in at least 5 control and 5 ALL samples by western blot.

Response: We appreciate the reviewer’s comment about validating the mRNA expression data at the protein level in at least 5 control and 5 ALL samples by western blot. However, we want to point out that we did confirm cell surface protein expression of the receptors in T cells, B cells, NK cells and monocytes by conducting flow cytometry on all 42 patients and 20 control subjects and the data has been reported in Figures 3 – 5. Flow cytometry helps in determining cell surface protein expression as opposed to the total protein content determined by western blot unless you do cell fractionation to determine membrane, cytosolic and nuclear fractions. In the context of NK cells and T cells, the receptor-ligand interactions are important and therefore we focused on determining the cell surface protein expression by flow cytometry instead of western blot. Since, our IRB protocol does not permit us to re-contact the patients it is not possible to get any more blood samples from them to conduct any additional western blot analysis.

Reviewer’s Comment: I agree with the Authors regarding primary sample. I think, the alternative of primary samples is cell lines. Please check it in different ALL and other cell lines/ EVB transformed cells. Authors also have the scope to check the expression in different cell derived from same iPSC (if possible).

6. Please provide patients' demographic data as a table for both RNA seq as well as newly collected samples showing the epidemiological and clinicopathological features.

Response: The RNAseq expression data was obtained from the St. Jude Pediatric Cancer genomic (PeCan) database where a total of 729 B-ALL subjects and 313 T-ALL subjects’ data was analyzed. It is practically impossible to show the patient demographic, epidemiological and clinicopathological features in a table for that large cohort. Besides that, the patient demographic, epidemiological and clinicopathological is not available on St. Jude’s PeCan portal. However, we have added a table to show the demographic, epidemiological and clinicopathological features for the 42 patients that were recruited in our cohort study.

Reviewer’s Comment: It is not the right type of answer. Moreover, authors are presenting wrong way of of Table1. Please see the paper (https://doi.org/10.1002/ijc.23834) how it present hundreds of patient’s clinicopathological data. In this way, clinicopathological features of several millions of patient could be presented in a small table.

7. In recent discoveries, Stress is the most important factor, which could induce tumorigenesis and promote cancer development via modulation of genetic and epigenetic changes. Stress has the potential power to generate almost all the hallmarks of cancer in a healthy person. Moreover, it could modulate the chemotherapeutic efficacy via the regulation of differential gene expressions in different types of cancer (https://doi.org/10.3389/fonc.2020.01492; https://doi.org/10.1038/s41568-021-00395-5). In this regard, it will be more logical to see the expression of those genes in different types of immune cells and stratified by the stress level of the patients before and after chemotherapy.

Response: We completely agree with the reviewer’s comments about stress being an important factor for inducing tumorigenesis and promoting cancer development via modulation of genetic and epigenetic changes. In the context of leukemia, there have been studies that have reported occupational hazards and exposure to organic solvents like benzene to cause leukemia. Further studies are warranted to examine the role of these stressors in modulating the expression of immune receptors and their role in the induction and progression of leukemia in a pediatric population.

Reviewer’s Comment: Acceptable logic.

Author Response

1.      Author analyzed protein expression in different types of cells. It would be great if they could analyze mRNA expression in the same type of cells instead of PBMC at least in 5 normal and 5 patient subjects.

Response: We highly appreciate the reviewer’s comment and ideally would have analyzed the mRNA expression in different cell types instead of PBMC from the ALL patients. But one of our biggest limitation was the sample volume which is a great challenge when you work with pediatric patients. Our day 0 sample collection would take place simultaneously with all other blood draws for diagnostic purposes to identify whether the patient has leukemia or not. At that moment the child and the family have a lot of emotional, psychological and physical stress. We would consider ourselves lucky if we got 5 mls of blood from the patient. Considering all of those limitations we just could not isolate individual cell types and run mRNA qRT-PCR analysis on them. Therefore, we decided to do flow cytometry which would provide valuable information about the cell surface expression of the receptors which is very important for the functional outcome of the receptor-ligand interactions especially in the context of NK cell function.

Reviewer’s Comment: I agree with the Authors regarding primary sample. I think, the alternative of primary samples is cell lines. Please check it in different ALL and other cell lines/ EVB transformed cells. Authors also have the scope to check the expression in different cell derived from same iPSC (if possible).

Response to Reviewer’s comment: We greatly acknowledge the reviewer’s suggestion to use cell lines to validate the results that we have performed on primary cells obtained from ALL patients but using cell lines for validation comes with a caveat as cell lines are genetically manipulated this may alter their phenotype, native functions and their responsiveness to stimuli. Serial passage of cell lines can further cause genotypic and phenotypic variation over an extended period of time and genetic drift can also cause heterogeneity in cultures at a single point in time. Also, cultured cancer cells are almost certainly not representative of differentiated cells in the human body. Therefore, cell lines may not adequately represent primary cells and may provide different results. Also, using 3 or 4 cell lines may not adequately represent our ALL patient population and it will be really hard to adequately interpret the results due to these confounding factors. Furthermore, mRNA expression may not always result in protein expression. So, we request the reviewer to accept our reasoning for this comment. 

  1. In the introduction (last papa) describes the source and method of sample collection. Please relocate the para in the method section.

Response: We have removed the last paragraph from the introduction where the source and method of collection was described. The methods section already has all of the details of sample collection.

Reviewer’s Comment: I agree. Thank you.

Response to Reviewer’s comment: Thank you 

  1. In RNA seq data, generally several hundreds of genes are identified as deregulated genes. From the list of those genes, how authors prioritized a few selected genes? Please clarified it in the result and method section.

Response: It is true that RNAseq data can identify hundreds of genes that are either upregulated or downregulated but for the purpose of this study we queried the major activating and inhibitory receptor/ligands in the St. Jude Pediatric Cancer genomic (PeCan) database. As mentioned in the introduction section, three of the five receptors that were selected i.e. 2B4, CS1 and LLT1 were cloned, identified and characterized in our laboratory and so we were interested to further examine the relevance of these receptors in a pediatric ALL population and compare their expression with the NCR receptors: NKp46 and NKp30 that has been widely studied in ALL research. Therefore, these 5 receptors/ligand were chosen for this study.

Reviewer’s Comment:  I agree.

Response to Reviewer’s comment: Thank you 

  1. Authors only identified the deregulation of 5 genes. What is the mechanism of upregulation or downregulation in ALL? Please see some mechanistic aspects. Is it genetic or epigenetic regulation?

Response: This is an excellent question and we thank the reviewer for pointing that out. Specifically, in ALL we do not know the exact mechanism of either upregulation or down regulation of these receptors but the possibility of either genetic or epigentic regulation cannot be ruled out. Further mechanistic studies are warranted to ascertain the exact mechanism of action which is beyond the scope of this paper but would be a plausible next step for this project.

Reviewer’s Comment: If so, author could show some mechanistic functional aspects of upregulation of the said genes in ALL cell line (s).

Response to Reviewer’s comment: Again we would like to thank the reviewer for posing a very pertinent question but would like to state that showing mechanistic functional aspect is the logical next step for this project and it is beyond the scope of this paper. As the title itself indicates we have focused on the expression of the immune receptors in this paper. There are a lot of different angles from which we could study the mechanism of upregulation or downregulation of the genes in ALL and that itself would be a totally separate project. So, we request the reviewer to accept our reasoning for this comment. 

  1. Authors check the expression at the mRNA level. Please validate your mRNA data at the protein level in at least 5 control and 5 ALL samples by western blot.

Response: We appreciate the reviewer’s comment about validating the mRNA expression data at the protein level in at least 5 control and 5 ALL samples by western blot. However, we want to point out that we did confirm cell surface protein expression of the receptors in T cells, B cells, NK cells and monocytes by conducting flow cytometry on all 42 patients and 20 control subjects and the data has been reported in Figures 3 – 5. Flow cytometry helps in determining cell surface protein expression as opposed to the total protein content determined by western blot unless you do cell fractionation to determine membrane, cytosolic and nuclear fractions. In the context of NK cells and T cells, the receptor-ligand interactions are important and therefore we focused on determining the cell surface protein expression by flow cytometry instead of western blot. Since, our IRB protocol does not permit us to re-contact the patients it is not possible to get any more blood samples from them to conduct any additional western blot analysis.

Reviewer’s Comment: I agree with the Authors regarding primary sample. I think, the alternative of primary samples is cell lines. Please check it in different ALL and other cell lines/ EVB transformed cells. Authors also have the scope to check the expression in different cell derived from same iPSC (if possible).

Response to Reviewer’s comment: Again, we greatly appreciate the reviewer’s suggestion to use cell lines to validate the results that we have performed on primary cells obtained from ALL patients but using cell lines for validation comes with a caveat as cell lines are genetically manipulated this may alter their phenotype, native functions and their responsiveness to stimuli. Serial passage of cell lines can further cause genotypic and phenotypic variation over an extended period and genetic drift can also cause heterogeneity in cultures at a single point in time. Moreover, cultured cancer cells are almost certainly not representative of differentiated cells in the human body. Therefore, cell lines may not adequately represent primary cells and may provide different results. Also, using 3 or 4 cell lines may not adequately represent our ALL patient population and it will be really hard to adequately interpret the results due to these confounding factors. Furthermore, mRNA expression may not always result in protein expression. We have already shown cell surface protein expression on the ALL patient samples by flow cytometry which is a universally accepted method. So, we request the reviewer to accept our reasoning for this comment.

  1. Please provide patients' demographic data as a table for both RNA seq as well as newly collected samples showing the epidemiological and clinicopathological features.

Response: The RNAseq expression data was obtained from the St. Jude Pediatric Cancer genomic (PeCan) database where a total of 729 B-ALL subjects and 313 T-ALL subjects’ data was analyzed. It is practically impossible to show the patient demographic, epidemiological and clinicopathological features in a table for that large cohort. Besides that, the patient demographic, epidemiological and clinicopathological is not available on St. Jude’s PeCan portal. However, we have added a table to show the demographic, epidemiological and clinicopathological features for the 42 patients that were recruited in our cohort study.

Reviewer’s Comment: It is not the right type of answer. Moreover, authors are presenting wrong way of of Table1. Please see the paper (https://doi.org/10.1002/ijc.23834) how it present hundreds of patient’s clinicopathological data. In this way, clinicopathological features of several millions of patient could be presented in a small table.

Response to Reviewer’s comment: We apologize for not completely comprehending the way the reviewer wanted the data presented in the table. We thank the reviewer for providing an example and we have formatted the table accordingly. We believe that the table for clinicopathological features are in the acceptable format now. With regard to the patient population from St. Jude Pediatric Cancer genomic database, the clinicopathological features for that cohort is not available on the public domain so we are not in a position to obtain and show that data. Hope this satisfies the reviewer’s comment and it is acceptable.  

  1. In recent discoveries, Stress is the most important factor, which could induce tumorigenesis and promote cancer development via modulation of genetic and epigenetic changes. Stress has the potential power to generate almost all the hallmarks of cancer in a healthy person. Moreover, it could modulate the chemotherapeutic efficacy via the regulation of differential gene expressions in different types of cancer (https://doi.org/10.3389/fonc.2020.01492; https://doi.org/10.1038/s41568-021-00395-5). In this regard, it will be more logical to see the expression of those genes in different types of immune cells and stratified by the stress level of the patients before and after chemotherapy.

Response: We completely agree with the reviewer’s comments about stress being an important factor for inducing tumorigenesis and promoting cancer development via modulation of genetic and epigenetic changes. In the context of leukemia, there have been studies that have reported occupational hazards and exposure to organic solvents like benzene to cause leukemia. Further studies are warranted to examine the role of these stressors in modulating the expression of immune receptors and their role in the induction and progression of leukemia in a pediatric population.

Reviewer’s Comment: Acceptable logic.

Response to Reviewer’s comment: Thank you

Reviewer 3 Report

Dear Authors, 

I have no further comments on the revised manuscript.

Best regards

Author Response

Thank you for accepting our response.

Round 3

Reviewer 1 Report

Response to the reviewer is nicely presented.

Author Response

Thank you for accepting our response.